# Functional and Compositional Changes in the Fecal Microbiome of a Shorebird during Migratory Stopover

Kirsten Grond,[a]* Artemis S. Louyakis,[a]§ Sarah M. Hird[a,b]

[a]Department of Molecular and Cell Biology, University of Connecticut, Storrs, Connecticut, USA
[b]Institute for Systems Genomics, University of Connecticut, Storrs, Connecticut, USA

Kirsten Grond and Sarah M. Hird These authors contributed equally to this work. Author order was determined by total effort towards the paper.

**ABSTRACT** Shorebirds migrate long distances twice annually, which requires intense physiological and morphological adaptations, including the ability to rapidly gain weight via fat deposition at stopover locations. The role of the microbiome in weight gain in avian hosts is unresolved, but there is substantial evidence to support the hypothesis that the microbiome is involved with host weight from mammalian microbiome literature. Here, we collected 100 fecal samples of Ruddy Turnstones to investigate microbiome composition and function during stopover weight gain in Delaware Bay, USA. Using 16S rRNA sequencing on 90 of these samples and metatranscriptomic sequencing on 22, we show that taxonomic composition of the microbiome shifts during weight gain, as do functional aspects of the metatranscriptome. We identified 10 genes that are associated with weight class, and polyunsaturated fatty acid biosynthesis in the microbiota is significantly increasing as birds gain weight. Our results support that the microbiome is a dynamic feature of host biology that interacts with both the host and the environment and may be involved in the rapid weight gain of shorebirds.

**IMPORTANCE** Many animals migrate long distances annually, and these journeys require intense physiological and morphological adaptations. One such adaptation in shorebirds is the ability to rapidly gain weight at stopover locations in the middle of their migrations. The role of the microbiome in weight gain in birds is unresolved but is likely to play a role. Here, we collected 100 fecal samples from Ruddy Turnstones to investigate microbiome composition (who is there) and function (what they are doing) during stopover weight gain in Delaware Bay, USA. Using multiple molecular methods, we show that both taxonomic composition and function of the microbiome shifts during weight gain. We identified 10 genes that are associated with weight class, and polyunsaturated fatty acid biosynthesis in the microbiota is significantly increasing as birds gain weight. Our results support that the microbiome is a dynamic feature of host biology that interacts with both the host and the environment and may be involved in the rapid weight gain of shorebirds.

**KEYWORDS** 16S rRNA, fatty acid, metatranscriptome, migration, mRNA, Ruddy Turnstone, metatranscriptomics, shorebird

Address correspondence to Sarah M. Hird, sarah.hird@uconn.edu.

*Present address: Kirsten Grond, Department of Biological Sciences, University of Alaska Anchorage, Anchorage, AK, USA.

§Present address: Artemis S. Louyakis, Colgate-Palmolive Company, Piscataway, NJ, USA.

The authors declare a conflict of interest. Artemis S. Louyakis declares employment by private corporation Colgate-Palmolive Company; however, this company has had no influence or interaction with the contents of this manuscript.

Within every major vertebrate clade, there are species that undertake annual seasonal migrations to find optimal environmental conditions for survival or breeding: from lampreys to sea turtles to elephants to many species of bird. Migratory animals go through a myriad of physiological changes throughout their annual cycles to prepare for, accomplish, and recover from migration itself or to prepare for, accomplish, and recover from reproduction; for long-distance migratory birds, these changes associated with migration are the most extreme changes of their annual cycle (1, 2). Shorebirds (Order: Charadriiformes) undertake migrations of thousands of kilometers, twice a year, between their breeding grounds and nonbreeding grounds (e.g., Fig. 1). To prepare for migration, shorebirds absorb

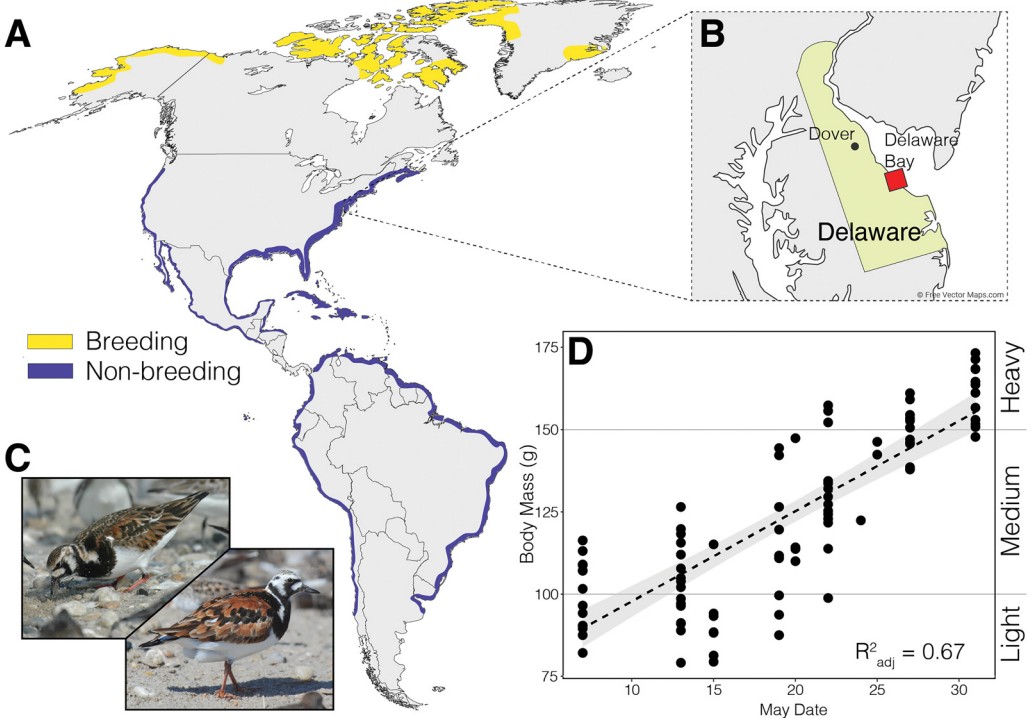

**FIG 1** Ruddy Turnstone attributes. (A) Distribution of North, Central, and South American breeding and nonbreeding grounds. (B) Sampling site (Delaware Bay, DE). (C) Female (left) and male (right) breeding plumage; photos (C) Gregory Breese, US Fish and Wildlife Service. (D) Weight distribution of 91 Ruddy Turnstones sampled during stopover in 2018, linear regression line with region of standard error highlighted in gray (adjusted $R^2$ = 0.67).

part of their digestive tract, which is unused during flight, to reduce body weight. They also increase the size of their pectoral muscles to maximize flight performance (3, 4). Prior to migration, shorebirds rapidly gain weight to fuel their flights, often almost doubling their body mass in as little as 14 days (5). This rapid weight gain is largely due to a short period of extreme foraging behavior, called hyperphagia. Birds increase their food intake by 20 to 40% and can accomplish a 7% mass gain per day (6, 7). The weight is largely comprised of fat, which provides the most efficient fuel to complete their migrations. Hyperphagia may not be the only mechanism responsible for shorebird weight gain; e.g., changes in the nutritive quality of dietary items during stopover or physiological changes to the intestinal barrier as birds regrow their organs may also contribute.

The role of the microbiota (the microorganisms that live on and inside a host) in the anatomical and physiological changes associated with vertebrate weight gains and losses of migration is a major outstanding question with both basic and applied implications. A vertebrate's microbiome is intimately involved in many aspects of vertebrate biology, including development, immunity, behavior, and digestion (reviewed in reference 8). The microbiome is associated with weight gain, with an applied focus on humans and model organisms (9, 10). Bacteria within the phylum Firmicutes have been associated with obesity when paired with a high fat diet ([11]; but see reference 12); conversely, other bacteria (notably *Bacteroidetes*) are associated with lean or normal weight (13–15). Obese mice with microbiomes rich in Firmicutes extract more energy from a given amount of food than lean mice with microbiomes relatively lower in Firmicutes (13). Strong mechanistic links between the microbiome and fat deposition involve *bacterial* metabolites, host gene regulation, and lipogenesis (16). The gut-microbiome-brain axis posits additional ways that the microbiome can influence weight in hosts, through food-seeking behavior, appetite, taste, and food preferences (reviewed in reference 17). Similar to the natural, high-fat state of shorebirds before migratory flight, many species of mammals rapidly gain weight before hibernation, which sustains them through periods of low food intake. For example, arctic ground squirrels (*Urocitellus parryii*) hibernate for 6 to 9 months and rely completely on the

fat mass accumulated during their active season. Although the role of the microbiome in fattening is unknown in this species, the microbiome was recently shown to be involved in maintaining lean mass during hibernation (18). The brown bear (*Ursus arctos*) is another hibernator whose microbiome undergoes annual changes: during the summer, while the bears are gaining weight, the microbiome is relatively higher in Firmicutes, which can increase fat deposition under controlled conditions when transplanted into germfree mice (19).

Fatty acids play important roles in fat deposition and in growth of muscles. In shorebirds, fatty acids are the primary energy source for long-distance migration because they are twice as energy dense as alternative sources of energy (e.g., carbohydrates, protein) and relatively anhydrous, meaning the birds do not need to carry water weight to access the energy during flight (20). Migratory shorebirds use diets high in specific lipids to enhance the performance of flight muscles and they are able to modify dietary fatty acids for storage (21). Fatty acids that are necessary for survival but not produced by the animal are "essential" and must be obtained through diet. During stopover at Delaware Bay, Ruddy Turnstone gut contents are 80 to 90% Horseshoe crab eggs, with the remaining 10 to 20% being "detritus" and "sand" with no identifiable insects or worms (22). (Notably, when not at stopover, Ruddy Turnstones have a wide dietary breadth, including arthropods and small crustaceans, mollusks, bird eggs, and carcasses.) The Ruddy Turnstone's specialized diet of horseshoe crab eggs at Delaware Bay is rich in lipids comprised of 16- (stearic acid) and 18-carbon (palmitic acid) saturated fatty acids and 18:1 monounsaturated fatty acid (oleic acid) (20). horseshoe crab eggs are also rich in essential n-3 and n-6 polyunsaturated fatty acids (PUFAs) (23). Monounsaturated acids, and specifically 18:1 oleate, may be preferable in some endurance scenarios as they offer a compromise between energy density and accessibility, whereas n-3 PUFAs are linked to enhanced performance in endurance exercises, possibly due to increased membrane fluidity accelerating transmembrane lipid transport (21).

In wild shorebirds, the increased food input combined with internal physiological shifts raises the question of how the microbiome affects and is affected by these changes. The taxa within the microbiome are frequently decoupled from the functional potential of the community (e.g., [15]) with dramatic differences in variance of these two metrics. Questions remain about the relative roles of taxa and function in community assembly and host-microbe interactions. Understanding how taxonomic composition and microbial function are related to a host's physiological changes may reveal the mechanisms used in weight gain before migratory flight, specifically, and in adapting to changing environmental conditions, more broadly.

The compositional dynamics of the microbial taxa in the gut microbiome of shorebirds has been studied using 16S rRNA gene amplicon sequencing at different stages of the annual cycle. Geographic location is significantly correlated with the taxonomic composition of the microbiome across multiple host species (24), but in highly similar environments, bacterial community structure can be host species specific (25). After hatch, shorebird gut communities grow exponentially for 2 days and then stabilize (26). Long-distance migrations impact the composition and community dynamics of the shorebird microbiome (27, 28). These 16S rRNA-based studies have robustly characterized taxonomic dynamics but, by definition, cannot describe the functional aspects of the community. Bacterial communities can assemble based on function (29), and phylogenetic and functional diversity can reveal unique aspects of microbial communities (30). Because bird microbiomes are generally poorly described (e.g., low resolution, unidentified taxa), empirical genomic data are essential. Metatranscriptomics complements 16S rRNA amplicon data by extracting the total RNA in a sample, then enriching for and sequencing the mRNA to identify the recently transcribed genes. Transcriptomics can reveal patterns at a relatively fine scale, resulting in a deep characterization of the active processes, such as pathway utilization and unexpected gene activity. A comprehensive investigation into the taxonomic composition and functional expression of the microbiome in wild migrating birds will provide insight into how microorganisms relate to weight gain during migration, a physiologically critical period in a vertebrate. Furthermore, because body condition impacts breeding success and performance of migrating shorebirds (31), understanding the contributing factors of successful

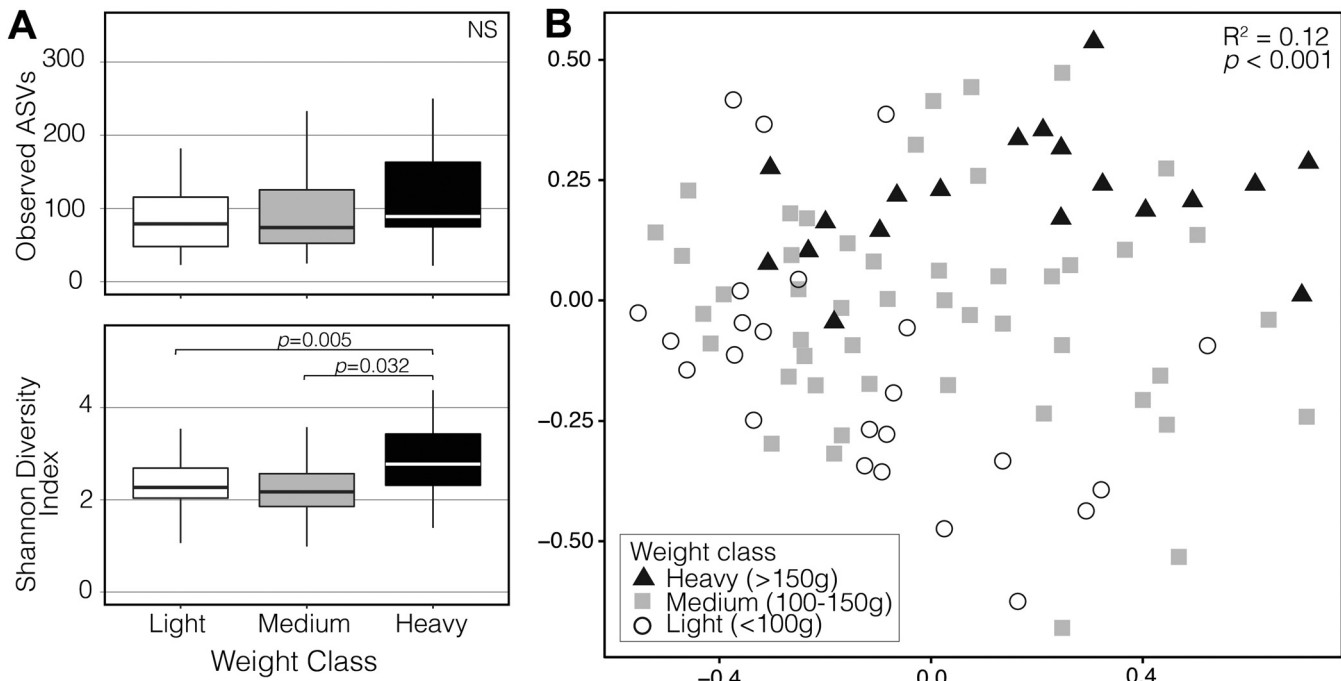

**FIG 2** Alpha diversity of the three weight classes (*P*-values < 0.005 shown). (B) Nonmetric Multidimensional Scaling ordination constructed from a Bray-Curtis matrix of 16S rRNA gene communities. Shapes and colors represent the three weight classes (black/triangle = Heavy, gray/square = Medium, white/circle = Light).

weight gain and migration have conservation implications for shorebirds, whose populations have shown a general and global decline in recent years (reviewed in reference 32).

We investigated the Ruddy Turnstone (*Arenaria interpres*) fecal microbiome, collected in Delaware Bay at different stages of fattening during the spring-migration staging period, using both 16S rRNA gene amplicon and metatranscriptome sequencing. Ruddy Turnstones stop in Delaware Bay for approximately 2 weeks during the month of May on their northwards migration and double their body mass during this time period by intensive foraging on the eggs of the horseshoe crab, *Limulus polyphemus* (22). Female Ruddy Turnstones are slightly bigger than males (Fig. 1C) and the two sexes often initiate migration at different times. The primary aims of our study were (i) compare taxonomic and functional microbiome profiles in shorebird microbiomes during weight gain at stopover, and (ii) identify significantly differentially expressed microbial genes to elucidate microbial pathways important to changing host weight and potential variation between sexes. We expect to see changes in both taxonomic and functional microbial communities because of the bird's rapid weight gain and its associated physiological changes. In addition, we predict an increase in gene expression of pathways related to fatty acid metabolism as a mechanism to facilitate said weight gain.

## RESULTS

Ruddy Turnstones consistently gained weight over the 2018 mid-migration stopover at Delaware Bay (Fig. 1; Linear Regression Model, $F_{1,89} = 182.8$, adj. $R^2 = 0.67$, $P < 0.001$). Body mass significantly differed among our three sampling sites (ANOVA: $F_{2,88} = 78.51$, $P < 0.001$), with the lowest bird weights at Swains and the highest weights at Back North (Fig. S1).

From the 100 fecal samples collected from Ruddy Turnstones, 90 samples were of sufficient postextraction quality for 16S rRNA gene sequencing. We sampled 45 female (F) and 41 male (M) Turnstones. Four individuals could not reliably be assigned a sex and are referred to as unknown (U); birds belonged to the following weight class sample sizes: light (N = 34), medium (N = 28), and heavy (N = 28). After quality control, we retained 3,700,042 high quality sequences across the 90 samples with an average of 40,660 ± 1,733 SE sequences per sample.

**Taxonomic composition and diversity using 16S rRNA. (i) Alpha diversity.** Shannon's diversity index (here "Shannon") significantly differed among weight classes (Fig. 2A, ANOVA: Shannon $F_{2,88} = 5.648$, $P = 0.005$). Light and medium weight birds differed significantly from

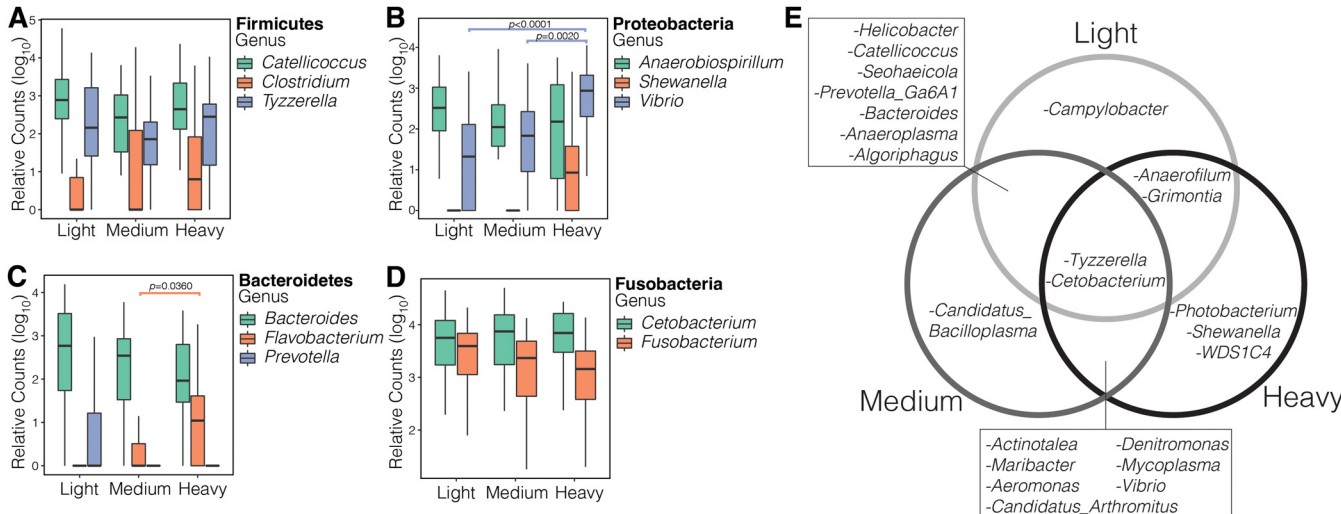

**FIG 3** Relative abundance of the dominant genera within the four most abundant phyla (A) Firmicutes, (B) Proteobacteria, (C) Bacteroidetes, (D) Fusobacteria. Genera are separated by weight on the *x* axis; all significant ($P < 0.05$) changes within the genera across weight classes are noted with respective *P*-values. (E) Genera detected by DESeq2 analysis to be differentially abundant in two pairwise comparisons (light v medium, medium v heavy, light v heavy). Because some genera had more than one ASV differentially expressed, and those ASVs might have been present in both categories, it is possible to have genera overexpressed in all three categories.

heavy birds, but not from each other (TukeyHSD: <100 g to >150 g, $P = 0.005$; 100 to 150 g to >150 g, $P = 0.032$; <100 g to 100-150 g, $P = 0.876$). Observed number of ASVs did not significantly differ between weight classes (ANOVA: Observed $F_{2,88} = 2.967$, $P = 0.057$). Alpha diversity significantly differed among sampling sites (ANOVA: Shannon $F_{2,88} = 6.449$, $P = 0.002$; Observed $F_{2,88} = 3.335$, $P = 0.040$), but did not differ between sexes (ANOVA: Shannon $F_{1,88} = 0.136$, $P = 0.873$; Observed $F_{1,88} = 0.493$, $P = 0.612$).

**(ii) Beta diversity.** The NMDS plot showed clustering in microbiome communities by weight class and showed directional change from light to heavy birds; medium weight class birds appeared to be located in between the light and heavy weight classes (Fig. 2B). PERMANOVA by weight class ($R^2 = 12\%$) and sampling site ($R^2 = 3.6\%$) were significantly associated with variation in microbiome composition ($P < 0.001$). Sex of the birds was not significantly correlated with microbiome composition (PERMANOVA: $F_{2,84} = 0.62$, $R^2 = 0.014$, $P = 0.994$). Homogeneity of variance (beta dispersion) did not significantly differ among weight classes (Permutest: $F_{8,84} = 0.75$, $P = 0.668$), sexes (Permutest: $F_{2,84} = 0.50$, $P = 0.612$), or sampling sites (Permutest: $F_{2,88} = 1.18$, $P = 0.330$). We were unable to separate the effects of weight and collection date due to the strong correlation between these variables. We compared the $R^2$ between models that included weight only, date only, and the date*weight interaction. All three variables were significant ($P < 0.001$) with the highest $R^2$ represented by the date*weight interaction (0.1685), followed by date (0.1588) and weight (0.1408).

**(iii) Community composition.** Twenty-six phyla were identified across all samples; five phyla comprised 97.3% of all sequences. The dominant phylum was Fusobacteria (40.7%), followed by Proteobacteria (26.7%), Firmicutes (18.9%), Bacteroidetes (5.8%), and Tenericutes (5.2%; Fig. S1). The 784 genera detected contained 4 463 ASVs. The Fusobacteria phylum was dominated by two genera, *Fusobacterium* and *Cetobacterium*, which together comprised >99.9% of sequences within this phylum. *Helicobacter* and *Campylobacter* were the dominant genera within the Proteobacteria, *Catellicoccus* within the Firmicutes, and *Bacteroides* and *Flavobacterium* within the Bacteroidetes. Within the most abundant genera, only *Vibrio* and *Flavobacterium* were significantly different between any weight classes (Fig. 3A to D; full statistical tests in Table S1).

**(iv) Differential abundance of taxa.** Pairwise tests identified many genera associated with weight class (Table S2). Those that were consistent across both pairwise tests (e.g., taxa significantly associated with light weight birds in both the light against medium and light against heavy comparisons) revealed 13 differentially abundant genera. Since some of

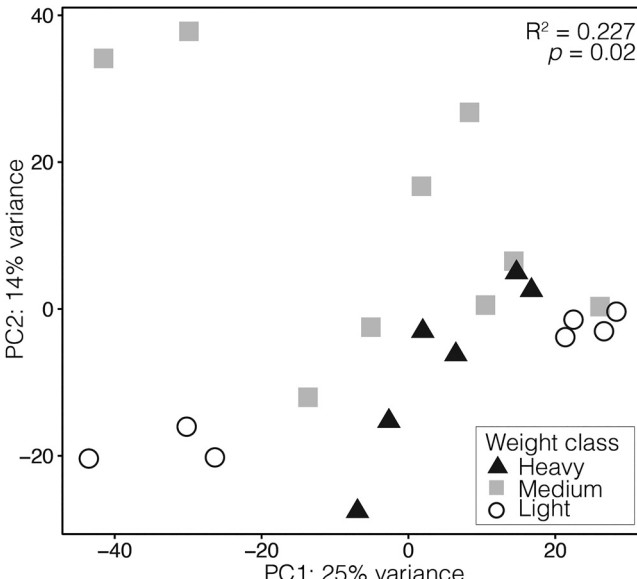

**FIG 4** Principal-component analysis of fecal metatranscriptomes collected from Ruddy Turnstones of different weights in Delaware Bay. Transcript counts were log-transformed, and colors/shapes represent the three bird weight classes.

these genera contained more than one differentially expressed ASV, two genera were differentially expressed in all three weight categories: *Tyzzerella* and *Cetobacterium* (Fig. 3E). In the light birds, only *Campylobacter* was differentially expressed in both its pairwise comparisons; in the medium birds, only *"Candidatus Bacilloplasma"*; and in the heavy birds, *Photobacterium*, *Shewanella,* and *WDS1C4*. Furthermore, *Vibrio* was overexpressed in the medium and heavy birds, compared to the light. *Helicobacter, Catellicoccus, Seohaeicola* and *Prevotella Ga6A1 group* were significantly higher in the light and medium birds compared to the heavy and *Grimontia* was significantly higher in the light and heavy birds compared to medium.

**Functional dynamics of the metatranscriptome.** After ribodepletion and library preparation, 22 out of 40 initially selected RNA samples were suitable for metatranscriptome sequencing, including 14 males and eight females in the following weight class sample sizes: light (N = 7), medium (N = 9), and heavy (N = 5). A total of 143,039 unique transcripts, 30,057 UniProt IDs, and 6, 481 KEGG Orthologys were detected (before filtering low represented transcripts).

We detected a significant difference in the functional gene community (based on KEGG Orthology, or "KO") among the three weight classes (Fig. 4, PerMANOVA: $F_{2,19} = 2.78$, $R^2 = 0.227$, $P = 0.02$), but not between sexes (PerMANOVA: $F_{1,20} = 1.56$, $R^2 = 0.027$, $P = 0.605$). Light birds and medium birds differed significantly from each other ($P_{adj} = 0.018$), but no difference in functional community was detected between light birds and heavy birds ($P_{adj} = 0.366$) or between medium birds and heavy birds ($P_{adj} = 0.226$). PCA beta dispersion differed among weight classes (ANOVA: $F_{2,19} = 5.10$, $P = 0.017$), which was driven by a significant difference between light birds and heavy birds (TukeyHSD: $P_{adj} = 0.020$). Beta dispersion did not differ between light birds and medium birds (TukeyHSD: $P_{adj} = 0.074$) or between medium birds and heavy birds (TukeyHSD: $P_{adj} = 0.588$).

**(i) Differential expression.** Several KOs were differentially expressed among weight classes (light versus medium, N = 7; light versus heavy, N = 4; medium versus heavy, N = 1; Fig. 5, Table S3). One, K06422, was differentially expressed between female and male individuals, as well as in light birds. K06422 is an unclassified gene that is active in small acid-soluble spore protein E (*sspE*) production during cell growth.

Expression of three polyunsaturated fatty acids were significantly associated with weight (Fig. 6): linoleic acid metabolism ($R^2 = 0.19$, $P = 0.024$), alpha-linoleic acid metabolism ($R^2 = 0.23$, $P = 0.014$), arachidonic acid metabolism ($R^2 = 0.39$, $P = 0.001$). Biosynthesis of unsaturated fatty acids was not significantly associated with weight ($R^2 = 0.03$, $P = 0.216$).

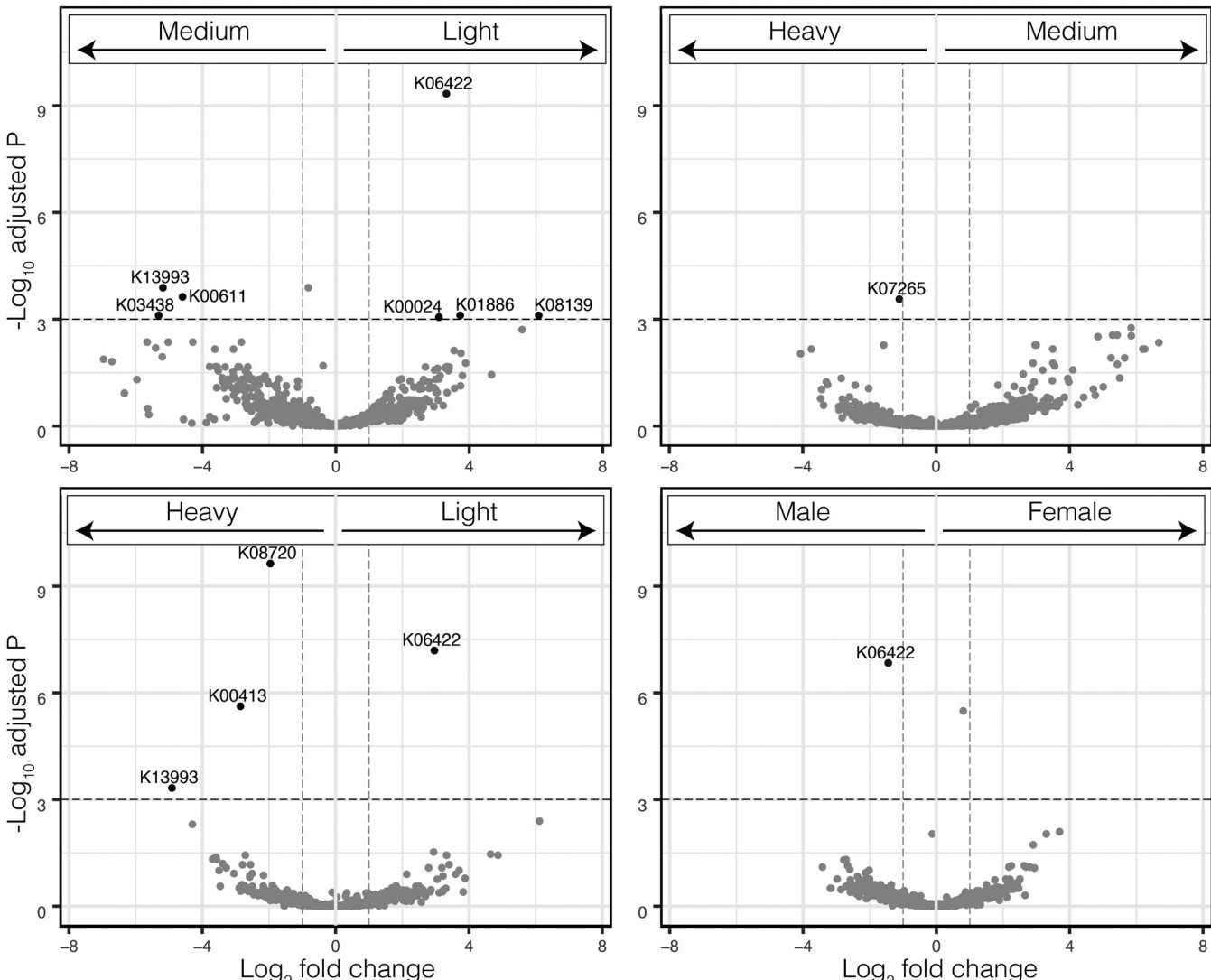

**FIG 5** Volcano plots showing differentially expressed genes in metatranscriptomes from different weight classes and sexes of Ruddy Turnstones. The horizontal lines represent $P = 0.001$. Genes that are differentially over or under expressed are identified with corresponding KEGG Orthology.

A clustering analysis was performed to group genes with shared expression patterns together, resulting in four clusters at $P < 0.05$ (Fig. 7, Table S4). Group 1 (N = 130 genes) and group 3 (N = 76 genes) had light birds with contrasting expression patterns to the medium and heavy birds. Group 2 (N = 9 genes) and group 4 (N = 22 genes) had medium weight birds as the highest or lowest expression group, respectively. All genes, regardless of significance, were analyzed as well and resulted in the same four expression patterns.

## DISCUSSION

The microbiome is intertwined with host health, behavior, and fitness, and the microbiota can play a key role in weight gain in model organisms and humans (11, 33). Shorebirds participate in extreme foraging behavior and rapid weight gain at stopovers during long distance migrations. Examining extreme behaviors or physiological processes may provide novel insight into the limits of how microbes facilitate vertebrate biology. In a broader sense, many animals experience periods of rapid or sustained weight gain (e.g., before migration or hibernation) or periods of weight loss, fasting, or starvation that must be recovered from. Here, we have shown in a system with highly similar diets across populations (22) at geographically close sampling sites, that as the host's body undergoes rapid weight gain, the taxonomic

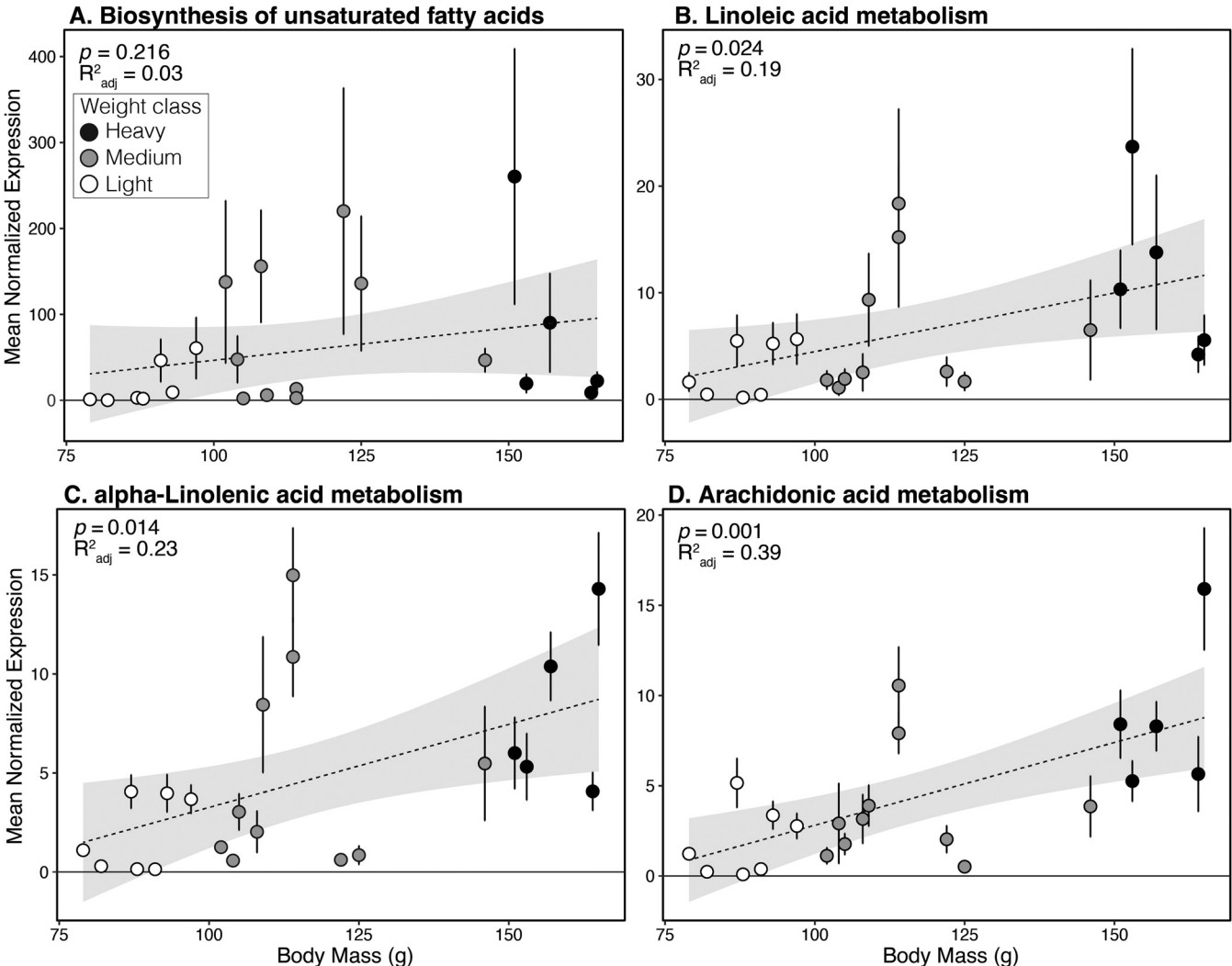

**FIG 6** Linear regression (dotted line with gray 95% confidence intervals) of mean normalized expression of (A) Biosynthesis of unsaturated fatty acids, (B) Linoleic acid metabolism, (C) alpha-Linoleic acid metabolism, (D) Arachidonic acid metabolism. Error bars represent standard errors.

composition of the fecal microbiome changes in tandem with the host, as do some of the functional capabilities of the microbiome.

Migration causes a significant disturbance to a bird's physiology and homeostasis; individual migrants can lose diversity of the microbiome during flight (34) and a corresponding successional recovery of the microbiota makes intuitive sense. Ruddy Turnstones gain an average of 50% body weight during stopover at Delaware Bay; therefore, the weight of a bird is a proxy for how long it has been at stopover and we can use this information to investigate changes through time. The three weight classes, light (birds that have most recently arrived at stopover), medium (birds that are midway through stopover), and heavy (birds that have been at stopover the longest and may depart soon), form significantly distinct clusters in our beta diversity ordinations based on 16S taxonomic composition. Furthermore, the medium weight birds are generally distributed between the low and heavy clusters (Fig. 2). This pattern—where weight classes are unique and progress from light to medium to heavy—implies successional change in the taxonomic composition of the fecal microbiome during stopover.

We used two methods to identify patterns within the taxa of the microbiomes: differential abundance analysis and ANOVA of the most abundant genera within the most abundant phyla. Succession within the microbiota may be further corroborated by the differentially abundant genera found across multiple pairwise tests (Fig. 3, Table S1). Of particular note is

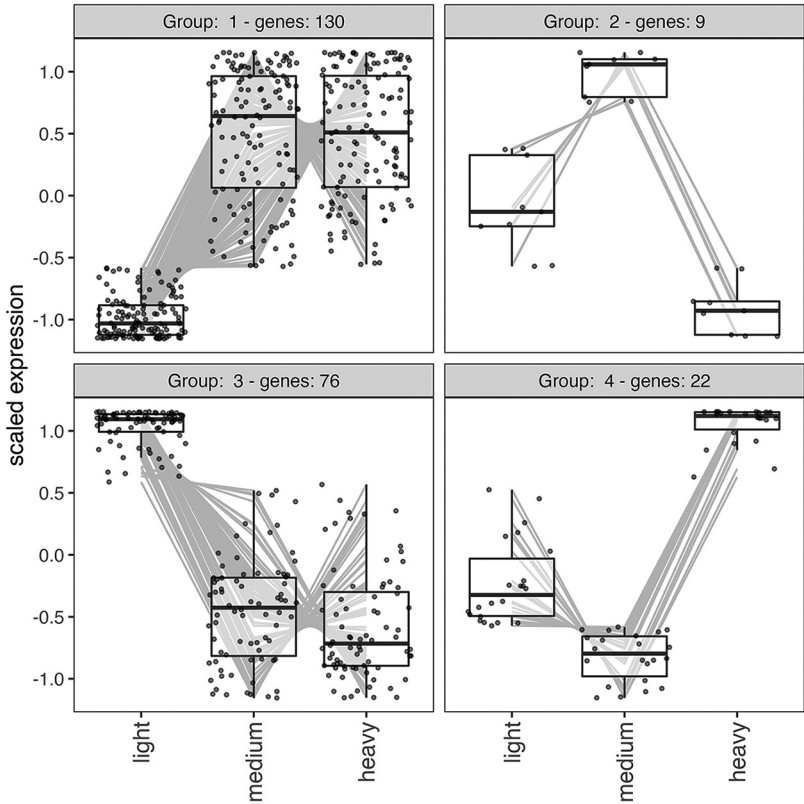

**FIG 7** Clusters of genes with significantly similar expression across the weight classes (alpha = 0.05). *y* axis represents Z-score. Positive values are upregulated compared to the average Z-score and vice versa for negative values.

that the genera *Helicobacter*, *Catellicoccus,* and *Campylobacter* are known bird gut-associated bacteria (35, 36) and these are more abundant in the light (or light and medium) birds. One hypothesis is that as the birds lose weight mid-migration (from declining fat reserves, muscle, and/or organ tissue), the bacteria that have coevolved to live with the birds are in the mucus that remains in the gut while all transient or less well adapted bacteria are flushed from the system or digested. After landing and as they gain weight, environmental (e.g., *Maribacter*, *Denitromonas*) and fatty acid producing marine bacteria (e.g., *Vibrio*, *Shewanella*, *Photobacterium* [37]) may be the first to colonize the postmigration community and flourish.

These taxonomic results contradicted our expectation that taxa from the Firmicutes would increase as birds gained weight, since Firmicutes have been associated with obesity or weight gain in mammals (but see reference 12). The most abundant Firmicutes in the shorebirds were not significantly different between the weight classes (Fig. 3A). The genera *Flavobacterium* (Phylum: Flavobacteria) was significantly higher in heavy birds compared to medium birds and *Vibrio* (Phylum: Proteobacteria) were significantly higher in the heavy birds compared to both the light and medium birds (Fig. 3). Potentially of note is that we used fecal samples in this study, which is a good, but imperfect, proxy for the avian gut microbiome (38). This was due to our inability and unwillingness to sacrifice birds for their gut contents. In shorebirds, different gut compartments contain specific microbial communities (39) so further testing is required to determine whether the compositional changes in the gut were closer to our expectations than the fecal samples. Succession in microbiomes is seen in many vertebrate systems (e.g., [26, 40]) and occurs after periods of microbiome disturbance (e.g., antibiotic treatment). Our birds' alpha diversity statistics trended upward the heavier they got, perhaps supporting a successional recovery of the microbiota. Continued investigation into the stability and resilience of microbiomes postmigration, and especially across years, would tell us how stable the shorebird microbiome is, on both short and long-term scales.

Sampling site was statistically associated with beta diversity in our 16S rRNA analyses, although with an effect size approximately one quarter of the weight class. We also detected a pattern between bird weight and the three sampling sites (Fig. S2). We hypothesize this is because sampling sites vary in food quality and better-quality sites are defended by larger birds. Broad scale associations between (food) quality of the sampling site and weight of the birds was observed, with lighter birds more frequently found on the lower quality sites, and heavier birds on the higher quality sites (41, 42).

As a community, and in contrast to the patterns displayed in taxonomic beta diversity (Fig. 2B), the medium weight functional communities were not obviously intermediate to the light and heavy weight categories (Fig. 4), although weight class was significantly associated with the variation of the samples ($R^2 = 0.22$, $P = 0.02$). Additionally, there appeared to be separation of the light birds into two clusters on PC1. These clusters did not correspond to any variable we could identify, including site, sex, or relative weight. To identify genes that may be associated with weight or weight gain, we used multiple methods: an unbiased pairwise significance test (Fig. 5), a cluster analysis (Fig. 7), and by specifically looking at pathways involved in polyunsaturated fatty acid biosynthesis (Fig. 6).

Using the pairwise significance test, most functions were not significantly different between the weight classes; only 10 functions (KOs, Table S3) rose above the significance threshold of $P < 0.001$. Of these 10 functions, K06422 was detected in three of the four comparisons, as significantly higher in light birds (compared to both medium and heavy birds) and as the only significant difference between males and females (higher in males). K06422 is associated with *sspE*, a small acid-soluble spore protein. As sporulation is a response to starvation in some bacteria, it may be that this protein is overrepresented in the light birds because they are underweight (i.e., below the species' average nonmigration weight), and the microbiota had entered a stress/starvation response. The other functions associated with the light birds, K01886 and K08139 are also generally associated with cell growth and metabolism. The functions associated with the medium weight birds were all potentially associated with proteins (Arginine biosynthesis, ribosome biogenesis, and heat shock proteins). K13993 is also associated with tissue remodeling and is significantly higher in both the medium and heavy birds compared to light. This result seems counterintuitive as fattening shorebirds are known to switch from protein recovery postmigration to fat deposition around medium weight (5). However, bacteria are known to digest dietary protein to produce secondary metabolites, such as amino acids, which could play a thus far unknown role in shorebird fattening (43). Additionally, some tissues and organs are grown throughout the stopover period in other shorebirds (44) and may influence the microbiota.

The pathways identified in the heavy birds are more diverse than those in the previous groups. They include functions that refer to organismal systems and human diseases (K00413) and structural proteins (K07625). Of particular note is K08720, which is associated with *Vibrio* biofilm formation, and may specifically involve iron balance (45). *Vibrio* was one of the taxa that was associated with heavier birds and is known to be a main microbial producer of polyunsaturated fatty acids (37). *Vibrio* are also found in the horseshoe crab microbiome (46), so the increase could be due to dietary intake and benefit the host. Hosts could also be internally filtering for *Vibrio*, or both processes could happen in tandem.

The clustering analysis that grouped genes based on shared expression patterns detected four clusters. The 237 genes within these four clusters may contain interesting targets for groups of genes that perform together and differentially depending on bird weight. Since birds go through a period of protein recovery and immune suppression when they first land (i.e., complete a leg of the migration) (47), genes in Groups 1 and 3 may be affected by those processes. The arrival of birds on land also returns the microbes to an environment of nonstarvation, and many of the genes in group 3 could be those involved in cell growth and division. Conversely, and if the birds do benefit from the metabolites or products of the microbiota, as medium and heavy birds are prioritizing weight gain, the genes in Groups 1 and 3 may be downregulated and upregulated for fat deposition, respectively. The genes in Groups 2 and 4 could be those responding to the immediate recovery of the microbiota and the bird's final preparations for flight; shorebirds both gain and lose

particular muscles and tissues in response to and preparation for flight, and how the microbiota respond to those changes requires further investigation.

We investigated an *a priori* hypothesis that polyunsaturated fatty acid biosynthesis would increase as the birds fatten. "Essential" fatty acids are those an animal needs but cannot produce; the diet of horseshoe crab eggs provides the essential fatty acids birds need and PUFAs in particular are an extremely efficient way to store energy that shorebirds use to power their migrations. Alpha-Linoleic acid is an n-3 PUFA, whereas Lineoleic acid and Arachidonic acid are n-6 PUFAs. These different categories of PUFA can have multiple and antagonistic effects (48). We hypothesize that the microbiome may also be producing essential fatty acids for the birds during weight gain. This hypothesis would require tracking experiments to confirm, but in our analyses, all three PUFAs (Linoleic acid, alpha-Linolenic acid, and Arachidonic acid) significantly increased as the birds gained weight (Fig. 6). General biosynthesis of unsaturated fatty acids did not significantly increase with weight gain, indicating possible opposite patterns in other unsaturated FAs. A next step in our study is to investigate the full spectrum of fatty acids to identify patterns in abundance with weight change and pursue mechanistic explanations.

It is important to note potential confounding factors affecting our study. For example, working with wild animals in their natural habitat limited our ability for sterile sample collection, which could lead to potential contamination. However, the use of extraction and presequencing PCR negative controls minimized this risk. We were also limited by the birds' natural movements and fattening schedules, which did not allow for randomization of our data collection, and we could not fully distinguish between the effects of weight and date. Another confounding factor in our results is that although we filtered out nonbacterial transcripts from the data set for genes that are highly conserved across domains, some of the transcripts may originate from the birds and there is no *ad hoc* way to confirm or deny those results. Finally, some of our data need further validation through additional research, such as the importance of the produced fatty acids in the host and/or microbe metabolism.

## MATERIALS AND METHODS

**Sample collection.** Fecal samples were collected from 100 Ruddy Turnstones (*Arenaria interpres*) from 7 to 31 May 2018 at three beaches in Delaware Bay, DE (Fig. 1). Birds were captured using cannon nets as part of the Delaware Shorebird Project, a program from the Delaware Department of Natural Resources and Environmental Control. Upon capture, birds were placed in individual boxes lined with 10% bleach-sterilized trays for up to 10 min. A mesh platform above the tray avoided contamination of fecal samples by the birds' feet; see reference 49 for detailed sampling description. Fecal samples were preserved in DNA/RNA shield (Zymo Research, Irvine CA) upon collection, and frozen at −20°C within 2 h of capture. After sample collection, weight and biometric measurements (wing length, head and bill length) were collected and birds were sexed and aged based on plumage characteristics (Fig. 1C).

Samples were sorted into three weight classes to increase sample size for statistical analysis. Birds were classified as light (<100 g), medium (100 to 150 g), and heavy (>150 g). The medium weight category starts at 100 g, as this is the average weight of Turnstones during the wintering period (50). Many (if not all) birds, when they first land at Delaware Bay, are below the average wintering period weight and thus classified as "light" weight. "Heavy" was defined as a 50% increase above the wintering period average. Because all birds lose weight during migratory flight and gain weight during stopover, the weight categories are approximate indicators of how long they have been at the stopover location and how soon they may begin the next leg of their migration.

**DNA extraction and sequencing.** DNA and RNA were extracted simultaneously using the ZymoBIOMIC DNA/RNA Miniprep kit (Zymo Research, Irvine CA), following the parallel extraction protocol. Extracted RNA and DNA were stored at −80°C until sequencing.

For the 16S rRNA gene sequencing, the V4 region of the 16S rRNA gene was PCR amplified and sequenced at the University of Connecticut Microbial Analysis, Resources, and Services facility, following the standard operating procedure. Quant-iT PicoGreen kit was used to quantify DNA concentrations, and 30 ng of extracted DNA was used as the template to amplify the V4 region of the 16S rRNA gene. V4 primers (515F and 806R) with Illumina adapters and dual barcodes were used for amplification (51, 52). PCR conditions consisted of 95°C for 3.5 min, 30 cycles of 30 s at 95.0°C, 30 s at 50.0°C, and 90 s at 72.0°C, followed by final extension at 72.0°C for 10 min. PCR products were normalized based on the concentration of DNA from 250 to 400 bp and pooled. Pooled PCR products were cleaned using the Mag-Bind RxnPure Plus (Omega Bio-tek) according to the manufacturer's protocol, and the cleaned pool was sequenced on the MiSeq using v2 2 × 250 base pair kit (Illumina, Inc, San Diego, CA). Both kit controls and presequencing PCR controls were included as "negatives" with the above procedures.

For the metatranscriptomes, total RNA was quantified, and purity ratios determined for each sample using the NanoDrop 2000 spectrophotometer (Thermo Fisher Scientific, Waltham, MA, USA). To assess RNA

quality, total RNA was analyzed on the Agilent TapeStation 4200 (Agilent Technologies, Santa Clara, CA, USA) using the RNA High Sensitivity assay following the manufacturers protocol. Ribosomal Integrity Numbers (RIN) were recorded for each sample. Total RNA samples (300 ng of Qubit quantified total RNA input) were prepared for prokaryotic transcriptome sequencing by first ribodepleting bacterial rRNA using the RiboMinus Transcriptome isolation kit, Bacteria (ThermoFisher Scientific, Waltham, MA, USA). Ribodepletion efficiency was analyzed prior to the start of library preparation on the Agilent TapeStation 4200 (Agilent Technologies, Santa Clara, CA, USA) using the RNA High Sensitivity assay following the manufacturer's protocol. Efficient ribodepletion is supported by the disappearance of the 16S and 23S rRNA peaks ($\sim$1,000 nt and $\sim$2,000 nt, respectively), with the sample's electropherogram trace now showing a smear of shorter molecules ($<$1,000 nt).

Purified ribodepleted RNA underwent library preparation using the Illumina TruSeq Stranded mRNA Sample Preparation kit following the manufacturer's protocol modification for purified mRNA as input (Illumina, San Diego, CA, USA). Libraries were validated for length and adapter dimer removal using the Agilent TapeStation 4200 D1000 High Sensitivity assay (Agilent Technologies, Santa Clara, CA, USA) then quantified and normalized using the dsDNA High Sensitivity Assay for Qubit 3.0 (Life Technologies, Carlsbad, CA, USA). Sample libraries of sufficient quality were sequenced (Illumina MiSeq; paired end $2 \times 75$ bp read length) with a sequencing depth targeted at 7 to 10 M total paired end reads/sample at the Center for Genome Innovation at the University of Connecticut.

**Sequence quality control, assembly, annotation, and mapping.** For the 16S rRNA gene amplicon data, during standard Illumina demultiplexing, sequences were quality checked and trimmed to remove adaptors and barcodes. The DADA2 (v. 3.11) pipeline in R (v3.6.0) was used to quality control and process the reads (53, 54); low quality read areas were removed following the DADA2 default parameters. Following assessment of error rates, paired-end sequences were merged, and potentially chimeric sequences removed. All unique sequences at greater than $1\times$ abundance were then labeled as amplicon sequence variants, or ASVs, for taxonomic analysis. Sequences were assigned to taxonomy using the Ribosomal Database Project (RDP) Naïve Bayesian Classifier with the Silva (v. 132) reference database (55, 56). Sequences identified as chloroplast or mitochondrial sequences were removed from the data set. A multiple-sequence alignment was performed using the Decipher (v. 2.0) package (57), and a phylogenetic tree was constructed with the package phangorn (v2.4.0 [58]). Likely sequence contaminants were identified and removed using the decontam (v1.4.0 [59]) package using the negative-control samples as contaminants.

The metatranscriptome sequences were trimmed using Trimmomatic (v0.35 [60]) with a threshold of Q5, and rRNA was removed using SortMeRNA (61). The remaining sequences were *de novo* assembled using Trinity (v2.2.0 [62]) with the following parameters: fastq assembly (left read file contained forward and unpaired reads), minimum contig length of 75 bp and normalized reads. Alignment was completed using Bowtie2 (v2.2.9 [63]) and RSEM (v1.2.7 [64]) estimation was used for counts of sample replicates. RSEM estimates were rounded to nearest integer, length corrected (transcripts per million method), trimmed mean of M-values (TMM) adjusted for normalized expression values (EdgeR v3.16.5 [65]), and batch corrected with ARSyNseq (in NOISeq [66, 67]). The metatranscriptome assembly was annotated using Trinotate (v3.0.1) using the complete pipeline (http://trinotate.github.io) and all eukaryotic contigs were removed before downstream analysis.

**Statistical methods.** A linear regression was used to confirm weight gain of Ruddy Turnstone over the stopover period. To assess microbiome taxonomic compositional change with weight gain, alpha and beta diversity measures were calculated for the 16S rRNA gene amplicons. For alpha diversity analyses, samples were rarefied to 10 995 sequences, which was the lowest sequence coverage of our samples. Two alpha diversity measures were calculated using phyloseq (v1.28.0 [68]): the observed number of ASVs and Shannon's Diversity Index (69). Statistical significances for differences in alpha diversity for sites, weight classes, and sexes were calculated using analysis of variance (ANOVA) testing. For beta diversity, three distance metrics were calculated to describe differences between samples: Bray-Curtis dissimilarity, weighted UniFrac and unweighted UniFrac (70). These distance matrices were used for nonmetric multidimensional scaling (NMDS). The relative contributions to the variation in microbiome composition of three variables (weight class, sex, and sampling site) were calculated using permutational multivariate analysis of variance (PERMANOVA) with the adonis2 function from the *vegan* package (v2.5.6 [71]). We tested for homogeneity of variance among weight classes and sexes using the betadisper function, also in vegan. Differential abundance in taxa between weight classes was conducted using DESeq2 package in R (v1.24.0 [72]). Significance was set at $\alpha = 0.001$.

We conducted a Principal Components Analysis on the TMM normalized expression counts and identified significance and relative contributions of weight class and sex in functional composition. A pairwise PERMANOVA using the pairwise.perm.manova function in the RVAideMemoire package (v. 0.9.75 [73]) with 1,000 permutations was used to assess significance between weight classes.

Differential gene expression between weight classes and sexes was conducted using *NOISeq* package in R (v1.24.0; [66]) with significance set to at $\alpha = 0.001$. Volcano plots were constructed from NOISeq data using the EnhancedVolcano (v1.2.0 [74]) and ggplot2 (v3.3.0 [75]) packages. To identify similar patterns of expression among differentially expressed genes, a cluster analysis using the DESeq2 (v1.24.0 [76];) and the DEGreport (76) packages was conducted. Likelihood Ratio Testing ($\alpha_{adj} = 0.001/0.01/0.05$) identified differences in expression across all weight classes and identified gene clusters across groups using the degPatterns function from DEGreport.

Because Ruddy Turnstones' main activity during stopover is acquiring fat mass, we focused on genes and pathways related to lipid metabolism. A variance stabilizing transformation was applied to count matrices using the vst function in DESeq2. To assess the relationship between body weight of Ruddy Turnstones and specific expressed pathways, normalized expression plots were constructed for biosynthesis of all fatty acids (FAs), and three essential polyunsaturated fatty acids (PUFAs): Arachidonic acid, Linoleic acid, and alpha-Linolenic acid.

**Conclusion.** Migration is a taxonomically widespread and physiologically demanding behavior that requires coordination of many organismal systems. The role of the microbiota in the processes that prepare and sustain animals during migration is poorly understood but may be important for energy storage and usage. Here, we have shown that the taxonomic community within the fecal microbiota of a migrating shorebird changes during the rapid weight gain of stopover and that some of the functions of the microbiota are also changing as birds fatten. The potential microbial origin and/or processing of specific fatty acids is an exciting future direction for research involving migration and the microbiome. Finally, as the threats against wild animals continue to increase and wild birds continue to have their populations threatened by human behavior, our work supports the important role of high-quality habitat for shorebird staging. The conservation of such areas—where birds can obtain the quantity and quality of dietary items needed to successfully migrate—should be a priority.

**Ethics statement.** Samples were collected with permission from the Delaware Division of Fish and Wildlife-Department of Natural Resources and Environmental Control (2018-WSC-031 to K.G.) and the Federal Bird Banding Permit [23332 to Delaware Division of Fish and Wildlife-Department of Natural Resources and Environmental Control].

**Data availability.** Sequence and metadata are available on Figshare (https://doi.org/10.6084/m9 .figshare.11337716.v1).

## SUPPLEMENTAL MATERIAL

Supplemental material is available online only.

**FIG S1**, TIF file, 0.4 MB.

**FIG S2**, TIF file, 1.6 MB.

**TABLE S1**, XLSX file, 0.01 MB.

**TABLE S2**, XLSX file, 0.01 MB.

**TABLE S3**, XLSX file, 0.01 MB.

**TABLE S4**, XLSX file, 0.04 MB.

## ACKNOWLEDGMENTS

We thank the Delaware Division of Fish and Wildlife and the Delaware Shorebird Project for allowing us to collect samples under their permits. This work was supported by the University of Connecticut (start-up funding to S.M.H.). Fieldwork of this project was funded, in part, through a grant from the United States Fish & Wildlife Service's State Wildlife Grant Program. This work does not represent the opinions of the State of Delaware, Delaware Department of Natural Resources & Environmental Control or Delaware Division of Fish & Wildlife. We also thank the editor and three reviewers for improving the manuscript. Our work was conducted on the traditional and unceded land and territories of the Lenape Mohegan, Schaghticoke, Mashantucket Pequot, Eastern Pequot, Golden Hill Paugussett, and Nipmuc Peoples (CT), and the and Nanticoke Nations (DE).

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
