## [Reviewer comments · mSystems]

Functional and compositional changes in the fecal microbiome of a shorebird during migratory stopover

Sarah Hird, Kirsten Grond, and Artemis Louyakis

Corresponding Author(s): Sarah Hird, University of Connecticut

Review Timeline:

Submission Date:	November 15, 2022
Editorial Decision:	December 1, 2022
Revision Received:	January 3, 2023
Accepted:	January 4, 2023

Editor: Suzanne Ishaq

Reviewer(s): Disclosure of reviewer identity is with reference to reviewer comments included in decision letter(s). The following individuals involved in review of your submission have agreed to reveal their identity: Brian K Trevelline (Reviewer #2)

Transaction Report:

DOI: <https://doi.org/10.1128/msystems.01128-22>

December 1, 2022

Dr. Sarah M. Hird
University of Connecticut
Molecular and Cell Biology
91 N Eagleville Rd, U-3125
MCB
Storrs, Connecticut 06269

Re: mSystems01128-22 (Functional and compositional changes in the fecal microbiome during pre-migratory weight gain in a shorebird)

Dear Dr. Sarah M. Hird:

Thank you for submitting your manuscript to mSystems. We have completed our review and I am pleased to inform you that, in principle, we expect to accept it for publication in mSystems. However, acceptance will not be final until you have adequately addressed the reviewer comments.

The authors have put in significant work to improve the manuscript, and only a few minor corrections have been suggested. I encourage the authors to make these minor changes and resubmit, at which point I feel that it will be ready to accept.

Preparing Revision Guidelines

Sincerely,

Suzanne Ishaq

Editor, mSystems

Journals Department
Reviewer comments:

Reviewer #2 (Comments for the Author):

I commend the authors on a job well done on this revised manuscript. The authors have satisfied my major concerns, and provided good justification for those comments not addressed. Therefore, I believe that this manuscript is ready for publication. I look forward to seeing this important study in print at mSystems.

Sincerely,
Brian Trevelline

Reviewer #3 (Comments for the Author):

Response to reviewers

Using 16S rRNA sequencing and metatranscriptomics, Grond et al. investigated variation in bacterial community composition and function across various weight classes of migratory ruddy turnstones during stopover in coastal Delaware. The authors suggest that weight gain and coincident changes in microbial communities indicate that the microbiome may play a role in the process of weight gain. A potential successional change in microbial taxa was detected, with medium-weight individuals clustering intermediately between low and high-weight individuals in a beta diversity ordination. Differentially abundant taxa were detected across each weight class, with previously known gut-associated bacteria being more abundant in light and medium weight class individuals. Ten genes (with functions related to cell growth, metabolism, tissue remodeling, etc.) were found to be differentially expressed in pairwise comparisons between weight classes. Clustering analysis detected four groups of genes with similar patterns of expression, highlighting genes that may function together in each weight class. The expression levels of several pathways related to lipid metabolism were found to increase across weight classes. This paper provides interesting insight into microbial community changes associated with the dramatic physiological process of weight gain during migration stopover. There are sections of the manuscript that would benefit from minor editing for clarity.

Minor issues

1. This study uses fecal samples as a proxy for processes occurring in the gut. Since it's known that fecal microorganism communities can be different than gut communities, it would be worth highlighting this potential discrepancy. It might also help explain some of the unexpected results (such as not seeing the expected increase in Firmicutes in the heavy weight class).
2. At the end of the introduction, outlining the expected outcomes of each of the study aims would be helpful to the reader.
3. Throughout the manuscript there are long sentences with lots of commas that are a bit difficult to read. Clarity could be improved by reorganizing and shortening these sentences.
4. The phrase "pre-migratory" is used throughout the manuscript, and it is at times unclear if the authors are referring to physiological changes occurring at the stopover sites or before the initiation of migration. Could a different term be used for each of these phases?
5. Line 28: Change "requires" to "require".
6. Line 68: Change "has" to "have".
7. Line 70: Should be "extract more energy from a given amount".
8. Paragraph at lines 85-103: Specifically mention that PUFAs are essential fatty acids. Essential fatty acids are defined, but it's unclear if the lipids present in Horseshoe crabs are essential fatty acids.
9. Line 99: Add (PUFA) after polyunsaturated fatty acids.
10. Line 117-199: The sentence that starts with "Long-distance migrations" is not worded clearly.

11. Line 456, 457: The phrasing "when they first land" and "immediate arrival" are a bit vague - is this referring to when they arrive at stopover sites? Or at the final destination of the migration?

Thank you for the comments on our manuscript; they have strengthened the final paper and we are grateful. Below are the specific changes we made to the manuscript to address reviewer comments (text from reviewer in black, our response in blue).

- Kirsten Grond, Artemis Louyakis, Sarah Hird

Response To Reviewers

1. This study uses fecal samples as a proxy for processes occurring in the gut. Since it's known that fecal microorganism communities can be different than gut communities, it would be worth highlighting this potential discrepancy. It might also help explain some of the unexpected results (such as not seeing the expected increase in Firmicutes in the heavy weight class).

We added a few sentences to the discussion addressing this potential problem: "Potentially of note is that we used fecal samples in this study, which is a good, but imperfect, proxy for the avian gut microbiome [66]. This was due to our inability and unwillingness to sacrifice birds for their gut contents. In shorebirds, different gut compartments contain specific microbial communities [67] so further testing is required to determine whether the compositional changes in the gut were closer to our expectations than the fecal samples."

2. At the end of the introduction, outlining the expected outcomes of each of the study aims would be helpful to the reader.

We added expected outcomes to the end of the introduction: "We expect to see changes in both taxonomic and functional microbial communities because of the bird's rapid weight gain and its associated physiological changes. In addition, we predict an increase in gene expression of pathways related to fatty acid metabolism as a mechanism to facilitate said weight gain."

3. Throughout the manuscript there are long sentences with lots of commas that are a bit difficult to read. Clarity could be improved by reorganizing and shortening these sentences.

We attempted to find difficult to read sentences but were generally happy with the readability of the manuscript and did not end up editing many sentences for this comment.

4. The phrase "pre-migratory" is used throughout the manuscript, and it is at times unclear if the authors are referring to physiological changes occurring at the stopover sites or before the initiation of migration. Could a different term be used for each of these phases?

For clarity, we have changed the wording for each use of pre-migratory.

5. Line 28: Change "requires" to "require".

Thank you, corrected.

6. Line 68: Change "has" to "have".

Thank you, corrected.

7. Line 70: Should be "extract more energy from a given amount".

Thank you, corrected.

8. Paragraph at lines 85-103: Specifically mention that PUFAs are essential fatty acids. Essential fatty acids are defined, but it's unclear if the lipids present in Horseshoe crabs are essential fatty acids.

Thank you, wording has been clarified.

9. Line 99: Add (PUFA) after polyunsaturated fatty acids.

Thank you, corrected.

10. Line 117-199: The sentence that starts with "Long-distance migrations" is not worded clearly.

Thank you, wording has been clarified.

11. Line 456, 457: The phrasing "when they first land" and "immediate arrival" are a bit vague - is this referring to when they arrive at stopover sites? Or at the final destination of the migration?

Thank you, wording has been clarified.

January 4, 2023

Dr. Sarah M. Hird
University of Connecticut
Molecular and Cell Biology
91 N Eagleville Rd, U-3125
MCB
Storrs, Connecticut 06269

Re: mSystems01128-22R1 (Functional and compositional changes in the fecal microbiome of a shorebird during migratory stopover)

Dear Dr. Sarah M. Hird:

Your manuscript has been accepted, and I am forwarding it to the ASM Journals Department for publication. For your reference, ASM Journals' address is given below. Before it can be scheduled for publication, your manuscript will be checked by the mSystems production staff to make sure that all elements meet the technical requirements for publication. They will contact you if anything needs to be revised before copyediting and production can begin. Otherwise, you will be notified when your proofs are ready to be viewed.

If you would like to submit a potential Featured Image, please email a file and a short legend to msystems@asmusa.org. Please note that we can only consider images that (i) the authors created or own and (ii) have not been previously published. By submitting, you agree that the image can be used under the same terms as the published article. File requirements: square dimensions (4" x 4"), 300 dpi resolution, RGB colorspace, TIF file format.

We recognize that the video files can become quite large, and so to avoid quality loss ASM suggests sending the video file via <https://www.wetransfer.com/>. When you have a final version of the video and the still ready to share, please send it to mSystems staff at msystems@asmusa.org.

Sincerely,

Suzanne Ishaq
Editor, mSystems

Journals Department
E-mail: mSystems@asmusa.org